# A Lesson from the Green Pass Experience in Italy: A Narrative Review

**DOI:** 10.3390/vaccines10091483

**Published:** 2022-09-06

**Authors:** Laura Leondina Campanozzi, Vittoradolfo Tambone, Massimo Ciccozzi

**Affiliations:** 1Research Unit of Bioethics and Humanities, Campus Bio-Medico University of Rome, 00128 Rome, Italy; 2Research Unit of Medical Statistic and Epidemiology, Campus Bio-Medico University of Rome, 00128 Rome, Italy

**Keywords:** Green Pass, COVID-19 pandemic, vaccine hesitancy, incentives, compulsory vaccination, discrimination, communication, ethics

## Abstract

The COVID-19 outbreak has raised several global challenges related to disease management while highlighting the need to embrace a multidimensional approach in dealing with events such as. Due to the singular features of SARS-CoV-2, an appropriate medical response was required to develop new vaccines able to tackle it effectively. Mass vaccination plans were thus promptly launched around the world. However, vaccine uptake has been coupled with growing concerns that have affected people’s willingness to get vaccinated. To promote compliance with vaccination campaigns, many governments introduced the use of vaccination certificates and immunization passports. Studies have discussed some benefits and cons coupled with the rollout of vaccine passports or certificates. This paper takes up and extends this discussion by showing the results of a mini- narrative review we undertook with the aim of critically summarizing the existing scholarly research on the Green Pass in Italy. In analyzing the 12 included records, we explored the scientific viability of this measure, as well as the concerns and criticisms it has raised and the recommendations that have been proposed to address them, as a starting point to consider how the lesson learned in the Italian context can contribute to informing future reflections and strategies in view ofanother pandemic event.

## 1. Introduction

The COVID-19 outbreak has raised several global challenges related to disease management while highlighting the need to embrace a multidimensional approach in dealing with events such as these, beyond the close implications of pandemics on public health and wellbeing [1]. The rapid spread of the SARS-CoV-2 and its variants has affectedmore than 520 million people worldwide, coupled with causing more than 6 million deaths [2], straining health care systems almost all over the world and compelling governments to enact an array of responses designed to contain its transmission and repercussions with little regard to the socio-economic costs [3]. School closings, movement and travel restrictions, social distancing, bans on public gatherings, and the use of facial masks and contact tracing were among the most common measures undertaken by governments to curb contagions [4,5]. Despite the provisions taken, a turning point in the course of the pandemic occurred with the advent of COVID-19 vaccines. Due to the singular feature of SARS-CoV-2’s ability to spread rapidly and easily, thus overloading healthcare systems and making clinical management difficult, it was immediately clear that SARS-CoV-2 appeared to be more dangerous and aggressive than the usual influenza viruses, necessitating an appropriate medical response and the development of new vaccines able to tackle it effectively [6]. This complex situation along with the news about the approval of the first vaccines for COVID-19 at the end of 2020 engendered optimism in society over the possibility of this pharmaceutical intervention to contain the spread of the virus [7]. Mass vaccination plans were thus promptly launched in several countries around the world aimed to reach the so-called herd immunity, while showing the high effectiveness of the new antidotes in stemming the incidence of SARS-CoV-2 infections [8,9]. Nevertheless, vaccine uptake has been coupled with growing concerns about their safety and efficacy [10], and these, along with the well-established phenomenon of vaccine opposition [11], have affected people’s willingness to get vaccinated and thus, the goal of herd immunity to keep the virus circulation under control. Accordingly, the World Health Organization has ranked vaccine hesitancy within the top ten of many threats to global health in 2019, underlining the support of health workers in order to provide trusted, credible information on vaccines [12,13].

In an attempt to promote compliance with vaccination campaigns toward reaching herd immunity while reducing the spread of the virus, several strategies were deployed worldwide [14,15]. Out of them, many governments, including Israel, the USA, Italy, Germany, and Switzerland, introduced the use of vaccination certificates and immunization passports, primarily as tools for dealing with those eager to restore normalcy to daily life, movement and the economy through the gradual relaxation of restrictions [16,17,18]. Studies have highlighted the lack of international consensus on the reasons for and format of these certificates, resulting in several models—physical as well as digital—being put in place, which are distinguished by their coverage (recovery, testing or vaccination) and by their official aim (domestic use, for example to travel within its own territories to access various social, cultural, and sporting events, etc.; or cross-border movement) [16,19]. As an example, on 14 June 2021, the EU Regulation introduced the so-called “EU COVID-19 certificate” to facilitate cross-border movement in the EU along with reducing the risk of disease transmission, envisaged for cured, vaccinated and negatively tested people [20]. The EU certificate should have played a coordinating role towards similar related health measures undertaken at the national level before or after its enforcement; however, the space to adopt further restrictions foreseen in the Regulation, the diversity of national public health rules on which these certificates would be grounded, and the different pattern of the epidemic curves in multiple countries has resulted in each member state introducing specificities in these certificates [15,19]. Regarding Italy in particular, the Government adopted a “Green Pass” certificate in July, 2021, requiring it to be displayed for access to a range of venues (restaurants, museums, sports and wellness centers, leisure and theme parks, cultural, social, and recreational centers, and game halls) and events (shows, festivals, conferences, and public competitions) [21]. Based on the broader use of the UE certificate, the Italian Green Pass was obtainable through vaccination, recovery from COVID-19, or a negative molecular or antigenic test in the previous 48 h. Thereafter, Italy enacted a set of decree-laws to implement and expand its use according to circumstances; until the Green Pass became required to enter workplaces and mandatory vaccination for those over 50 and for school and university staff was introduced [15,22]. Multiple studies have discussed some benefits and cons coupled with the rollout of vaccine passports or certificates, highlighting several relevant issues and featuring compelling considerations for policymakers [17,18,23,24,25,26,27,28,29]. This paper takes up and extends this discussion by showing the results of a mini- narrative review we undertook with the aim of critically summarizing the main existing scholarly research on the Green Pass in Italy and considering how the lesson learned in the Italian context can contribute to inform future reflections and strategies in view of another pandemic situation. Specifically, the review’s goals are to provide an overview of the strengths of the Italian Green Pass, in terms of its scientific validity, any concerns and criticisms raised by the program from different perspectives, and any proposals to address the issues it engendered. To the best of our knowledge, this is the first recap on this COVID-19 health measure with a focus on Italy, helping to outline emerging problems as well as recommendations related to this national program within the broader regulatory responsiveness to COVID-19 and preparedness for the next pandemic.

## 2. Materials and Methods

We performed a literature review on the Italian Green Pass policy in April 2022, with an update in early July. Since there was no previous existing review on this topic, before starting our literature search for publication, we identified some keywords, such as “*covid*”, “*vaccine*”, and “*green pass*”. Because these search terms were broad, to the point of including several items referring to similar programs both generally and in other countries, we then refined the search range in order to identify the studies relevant to the aims of our work. Therefore, we added the word “*Italy*” to the above ones, and combined them with the terms “*certificate*”, and “*passport*”, as our preliminary investigation showed that descriptions of the Green Pass policy are also associated with these words. These search terms were entered into three electronic databases, PubMed, Scopus, and the Web of Science (the Mach Search Terms used for the study are available in Appendix A). We only included publications written in English and Italian from peer-reviewed journals from 2011 onwards, taking into account that Italy adopted the green certificate in July 2011, with its introduction having been preceded by debate and any studies on the subject. This search resulted in 73 papers.

The first author began the literature search and excluded articles on the basis of duplication (n = 42). Then, the authors independently scanned the remaining 31 items by analyzing first the title and abstract and then the full-text manuscript to assess its eligibility. Concomitantly, the reference list of this sample was inspected to identify any additional relevant papers, as appropriate. In these phases, we included all records focused on the Italian Green Pass, specifically mentioning, describing and discussing its scientific viability, emerging concerns and criticisms related to this measure (also belonging to fields other than medicine), as well as feasible solutions to the challenges surrounding it. Papers that did not meet these inclusion criteria, that were not in English or Italian, focused in general on COVID-19 certificates or vaccine passports in general, or mentioned the Italian Green Pass without contributing significantly to the discussion upon the targets of this literature review, were removed. After a thorough evaluation and debate about few discrepancies on the identification of eligible articles, which were easily resolved between the authors, our final pool consisted of 12 publications (see Figure 1).

## 3. Results

Among the 12 records included in this narrative review, six papers explicitly focus on the Italian Green Pass experience, while the rest refer to this COVID-19 health measure in the context of a broader coverage on other issues somehow related to it, such as vaccine uptake and acceptance (n = 3), vaccine hesitancy (n = 1), Italian pandemic waves (n = 1), and the European Public perspective on vaccine mandates (n = 1). The results of reviewing the 12 chosen articles are presented in summary and were grouped into three main themes, which are not mutually exclusive. Consistent with our review targets, these are: (1) an *overview of the Italian Green Pass by its*
*scientific viability*; (2) *concerns and criticisms*; and (3) *proposed recommendations and alternatives*.

### 3.1. The Scientific Viability of the Italian Green Pass as Incentive-Based Model

Many references were made to Italy’s Green Pass approach as being an incentive-based model, showing potential in encouraging vaccine uptake, and allowing the decrease in virus circulation along with the gradual release of constraints. This topic has been approached from different perspectives in the included studies, mostly based on empirical investigations. Among the vaccination strategies, the Italian Green Pass is framed as an alternative approach to mandatory vaccination, which was adopted with the aim to promote immunization by prompting people to become vaccinated, as the recruitment trend was largely imperfect with respect to the goal of containing virus circulation at the time [29,30]. These points were restated in three more studies, where the Green Pass was compared to nudging strategies due to it being encouraging in incentivizing without imposing a decision [31], as a valid certificate is required to enter an increasing variety of venues and for a broad list of jobs, and was construed as a useful tool for achieving an overall change in behavior with the positive effects of wellbeing [32]. Among the reactions in Italy after the introduction of the mandatory Green Pass in Italy, Gallè et al. [33] pointed out that the sudden increase in vaccine reservations was consistent with the intent of this control measure to cope with the epidemiological emergency as well as to ensure safe social, economic, and work activities. These arguments are supported throughout the studies by official data coupled with a range of investigations. Stefanizzi et al. used open data from the Ministry of Health on the number of people who received the first dose of vaccination along with those who tested for SARS-CoV-2 before and after the introduction of the Green Pass policy to show the good effects of this health measure in increasing access to immunization [30]. However, the same data source shows a decrease in the number of vaccinations as well as a concomitant increase in the use of testing from September-October 2021. This leads authors to assume that the Green Pass is not a measure that is likely to incentivize vaccine skeptics and no-vaxxers who are more prone to resort to testing to obtain this certificate [30]. Designing a synthetic control model comparing the six countries that adopted COVID-19 certificates, including Italy, and using daily data on deaths, cases, vaccinations, and country-specific information, Mills and Rȕttenauer [34] also reported a positive relationship between the Green Pass and vaccine uptake in Italy before the announcement of this measure (from 6 August 2021 up until 8 November 2021), which was highest for people aged 18–29 years. Two other researchers noted similar tendencies with the positive impact of the Green Pass policy; in particular, regarding the pandemic waves in Italy, as shown by a time-trend analysis of official data, such as confirmed cases, deaths, hospitalizations, testing, outreach, and vaccination coverage [29]; on vaccine uptake, reducing adverse health outcomes; and on fostering the economic recovery in the short run, as appeared from estimates of the certificate’s contribution by using counterfactuals developed via innovation diffusion theory [35]. While no causality can be directly inferred from analyses of the estimates between the introduction of the Green Pass certificate and the positive effects mentioned above, although other restraining measures have also certainly helped in tackling the spread of the virus (e.g., physical and social distancing and the use of face masks), these studies agree on the importance of maintaining such nudge strategies as the Green Pass. Insofar as they are valuable in encouraging immunization compliance [29] and appear to be a more attractive and inclusive alternative to compulsory vaccination, leveraging more the benefits of vaccinating or testing rather than the punitive effects of not doing so [35]. The scientific viability of the Italian Green Pass was instead indirectly explored by Moccia et al. in a study aimed at investigating the phenomenon of vaccine hesitancy and any changes following the introduction of the mandatory Green Pass through online interviews [32]. Among the reasons mentioned by the sample behind their initial decision to become vaccinated, only 4.9% stated that they did so because it was the only way to obtain the Green Pass; whereas, “doing it for the Green Pass”, is mentioned as one of the drivers by those who initially decided not to become vaccinated [32] (p. 6), thereby confirming the assumption that this measure is an effective containment strategy. Therefore, it reduces the chances of interpersonal contact between individuals at higher risk of becoming infected, along with inducing behavioral changes, especially in those who are indecisive, fearful, uninformed, or reluctant [36].

### 3.2. The Green Pass and Its Detractors: Concerns and Criticisms

The introduction of the Green Pass in Italy was also coupled with protests and concerns by a segment of the population which refer to a variety of issues, such as privacy, discrimination based on one’s vaccination status, limitations on personal freedoms, expansion in the powers of control by the government, etc. Among the articles included in the final pool, three are focused on analyzing the nature and arguments underlying such opposition [31,36,37], four items instead mention this, but not as the main work target [15,34,37,38]. Spitale et al. have widely described the concerns of Green Pass opponents (widespread among university students) through the analysis of Telegram chats, a social listening tool on public health issues [31]. Although it has been found that the Green Pass argument acts as a catalyst for vaccine skepticism and, therefore, its detractors tend to share anti-vaccine views (especially those referring to the fear of possible side effects along with the lack of sound scientific evidence that vaccines work well) and conspiracy theories, the authors showed that Green Pass critics do not mainly resort to vaccine and conspiracy-related arguments to support their positions [31]. Rather, t most debate gravitates within the realm of legal aspects and the limitation of personal freedoms. These aspects have been overblown based on a widespread perception that the Green Pass was introduced as a nudging measure to avoid adopting (at least initially) mandatory vaccination. Thus a “cunning imposition”, as inferred from some chats, including the following: “I am against the green pass because I see it as a coercive and hypocritical tool put in place by the government because if they saw the vaccine as a safe way, they should have the consistency to make it compulsory and instead they don’t bother to do so” [31] (pp. 6–7). Two other studies have discussed or highlighted similar stances albeit from different perspectives. Palmieri and Goffin have provided an overview of Italian measures on compulsory vaccination through which they made it clear that a de facto obligation was already in place even before the government opted for a de jure obligation [22]. They focused in particular on Decree No. 172 of November 2021 from which two Green Passes begin to coexist: the enhanced Green Pass, also known as the Super Green Pass (issued to those vaccinated or recovered from COVID-19) mandatory to access an increasing number of places and activities, and the Basic Green Pass (released to those tested negative in the previous 72 or 48 h with a molecular or antigenic test, respectively), mandatory to access workplaces [22,39]. Through these measures, as noted by the authors, the pressure on the state has become higher towards vaccination, “imposing a sort of de facto compulsory vaccination for access to social life” [22] (p. 159). In assessing the acceptance of COVID-19 vaccination in a sample of elderly in southern Italy, Gallè et al., in turn, found an inverse relationship between being supportive of vaccination and fulfilling the questionnaire after the introduction of the Green Pass [33]. Namely, they observed a higher percentage of people not favoring vaccination in general and mandatory COVID-19 vaccination in particular, as well as the Green Pass, thereby confirming previous studies showing that the mandatory adoption of vaccine passports may affect negatively people’s motivation and willingness to become vaccinated by being perceived as a threat to human rights and civil liberties [33] (p. 9).

In this regard, most of the criticism of the Green Pass revolving around its being a discriminatory measure would have no scientific grounds [31]. In order to incentivize vaccination, the government initially extended the validity of the Green Pass beyond the deemed duration of vaccination coverage, thereby raising several concerns as well as a false sense of security in vaccinated people as they can become infected and infect others [15,37]. Second, and more importantly, the Green Pass would have worked to divide Italian society into two groups, the vaccinated and the unvaccinated, who see the latter having their fundamental rights (to work, study, and privacy) and personal freedoms (of opinion and movement) restricted [36,37]. These arguments are often backed through specific normative references, including the articles of the Italian Constitution no. 13 (establishing the inviolability of personal freedom), no. 32 (stating that nobody can be compelled to a particular health treatment except by a provision of law), and no. 120 (no one can hinder in any way the free movement of people or restrict the right to work in the national territory). In addition to the provisions of the aforementioned European Regulation no. 953/2021, which states to avoid any form of discrimination against people who have chosen to be unvaccinated, while the other urges states to issue certificates as economically as possible [15,36,37]. This last point has been mentioned to highlight a further controversial aspect of the Green Pass, as the Italian government has decided to exclude salivary antigen tests (which are less invasive and less expensive, although no less reliable) from the list of tests associated with the issuance of the green certificate [15] (p. 647). Tellingly, the detractors of the Green Pass have even bothered the Oviedo Convention in its art. no. 1 affirming the primacy of the human being over the mere interest of society and science. Against the backdrop of these normative frameworks, they stress that “it is unjust to protect life by limiting freedom” [31] (p. 8), and yet that that the individual harm from this measure is greater than the harm it would prevent to third parties [37]. Along this path, some have raised the concern that the Green Pass would advance a new concept of health as a social construct: this means, on the one hand, that the care of one’s own health will have a wider public value in the future, and on the other hand, that some freedoms will be subordinated and therefore lawfully limited when a danger to the health of others is at stake [38]. In light of the above, critics of the Green Pass view more broadly that the differential treatments introduced by this measure are unjust (and therefore discriminatory) from ethical, political-normative and scientific perspectives, hence it is an instrument of pervasive control over society, while representing a threat to democracy, and are in opposition as such [31,36,37]. Punctual analyzes of the concept of discrimination and the Italian measure on compulsory vaccination have shown, on the one hand, that the burden of argument falls on those who make the above-mentioned accusations against the Green Pass policy, insofar as they make a naive and unspecific use of this concept, in addition to misusing regulatory references [36,37]; on the other hand, interpretations of the normative references that opponents of the vaccine passport resort to are misplaced [22,36,37]. However, the social significance of these measures could convey discriminatory and stinging messages towards the unvaccinated, especially considering that it is easy to denigrate the choice of those who have decided not to vaccinate at the height of the pandemic, despite restrictive measures and strong pushes in that direction [37].

### 3.3. Proposed Recommendations and Alternatives

A range of recommendations and proposals about how to deal with the issues raised by and related to the introduction of the Green Pass in Italy has been considered in ten of the selected studies [15,29,30,31,32,33,34,35,36,37], also in view of any further pandemic events, including possible alternatives to measures targeting negative incentives. Specifically, proposals aimed at curbing the protests and concerns aroused by the Green Pass are mentioned only in three articles [31,35,37], while recommendations of a broader scope, although related to aspects pertaining to this measure, are covered to a greater extent [15,29,30,31,32,33,34,35,36,37]. Regarding the suggestions focused on the Green Pass, two studies emphasize the importance of adequately explaining the good reasons supporting the vaccine passport, disambiguating the purpose of this measure, which is to incentivize vaccination and thus protect all people [31,37]. Acknowledging the doubts of the Green Pass opponents, without rejecting their positions beforehand, critically deepening the concept of freedom that underlies opposition to it, offering more realistic conceptual models, and clarifying the regulatory basis of the Green Pass are other insights offered [31] (p. 11). Moreover, according to Oliu-Barton et al., governments’ policy decisions on such passports should take into account other aspects, including supply of vaccines and tests, political trust, international coordination and mutual acceptance of these certificates in order to prevent further divisions and inequities [35] (p. 5). Though acknowledging the scientific viability of the Green Pass, two studies have considered other opportunities or alternatives, such as introducing mandatory vaccination on the grounds that this measure does not seem to be able to intercept nonvaccine-skeptical groups [30], or looking at incentivizing vaccination with a payment that would compensate for taking a risk [37]. Regarding, the broader recommendations, in order to overcome barriers to vaccine uptake, studies suggest investing in resources for understanding the deep-rooted reasons behind vaccine hesitancy by engaging in a dialogue and active listening with the population, listening to their doubts and worries, and engaging especially with the most distrustful groups, as a tool for voicing such dissent, raising awareness among citizens, clarifying confusing aspects and building trust and confidence with institutions [15,31,32,34]. Health professionals and community representatives should be involved in this dialogue and participate in developing and disseminating mass media-based communication strategies to be both relevant and tailored to target people by using multiple languages, especially the most hesitant [32,33]. Unfolding effective education and information strategies is considered a key aspect during a health crisis such as a pandemic, mainly when accompanying the development of laws and policies aimed at gradually introducing mandatory measures [31,33] and coupled with planning interventions to support individuals’ autonomous motivation to become vaccinated [33], as strategies that build on personal persuasion are more in keeping with safeguarding personal freedom [15] (p. 647). In this regard, clearness, straightforwardness, accuracy and truthfulness should be unavoidable aspects of communication, when it involves providing data on the safety and efficacy of vaccines in particular [15,32,37], as a measure for increasing confidence in scientists, health professionals, and national and international authorities specializing in health issues [32,37]. On the educational side associated with the information process, Gometz suggested including elements of epistemological literacy in school and university curricula as a way to address the problem of widespread fake news in health care [36]. In order to deal more effectively with pandemics, Montanari-Vergallo et al. found it necessary to couple a solid, well-defined prevention strategy with vaccines and testing, focusing, for example, on reorganizing the health care system in general, strengthening territorial medicine, improving distance learning, etc. [15]. Indeed, the pandemic experience showed the importance of resorting to multifaced interventions (not only pharmaceutical) to control its spread [29]. Table 1 shows all the selected studies, with a brief description.

## 4. Discussion

This article has provided an overview of the literature about the Italian experience of the Green Pass, showing how this measure was mostly positively perceived as a public health instrument that could make an effective contribution to addressing specific challenges, namely that of incentivizing the population to become vaccinated to reduce virus outbreaks and gradually returning to the normalcy of daily life. Several studies have supported with empirical data the assumption of the scientific viability of the Italian Green Pass [29,30,32,34,35], reinforced by similar experiences in other countries [23,34,35,40]; although, we must be wary in claiming that there is a direct causal connection between the adoption of this measure and the previously mentioned benefits, which probably resulted from the concurrence of several factors according to a nonlinear logic that is well suited to interpret complex phenomena such as pandemic [41,42]. The pragmatic value of the Green Pass, however, coexists with the fact that the introduction of this policy has given rise to concerns mainly related to the mandatory restrictions imposed on individuals who had not been vaccinated or recovered from COVID-19, which, in turn, were perceived as not consonant with upholding human dignity and trust building with political and health authorities. Alemanno and Bialasiewicz have discussed these and other challenges posed by the introduction of similar programs, particularly the EU Digital COVID-19 Certificate, including the risk to create new forms of inequality through profiling mechanisms that separate safe people from unsafe people, or the illusion of pandemic safety, by ensuring citizens that through this certificate travel safety can be magically restored [17]. Comparable lessons have also been found in Israel’s COVID-19 Green Pass experience, since the adoption of this program along with the restrictions imposed on the unvaccinated and unrecovered was perceived as not in line with solidarity and trust-building, and thus as a violation of the rights of these people [43,44]. While these studies show similarities in the challenges raised by some forms of COVID-19 certificates probably due to the fact that they are emergency measures, they should lead us to be less critical of the specific Italian Green Pass program without minimizing its limitations and inaccuracies. Addressing issues surrounding the incentives and penalties, a previous study has shown that measures that provide penalties toward benefits, such as the Green Pass system, tend to be experienced as a violation of individual autonomy, in addition to providing external motivation for the desired behavior that will surely be less enduring as it is not supported by inner motivation [14]. This last aspect has importance in view of improving preparedness for the next pandemic whereby actions are not expected to be mechanically caused by external cues but are freely chosen. However, autonomy and free choices cannot occur where there is a failure to provide adequate information, and our review highlighted the moral duty to develop effective and transparent communication strategies as a preventive action to ensure sound and free behaviours while protecting vulnerable populations, not only in terms of health but also of their cultural background and values. In other words, the improvement of communication (and thus persuasion) techniques has to involve an ethical perspective in which truthfulness, caution, honesty, and a sense of circumstances would be a steady benchmark. Not least, the Italian Green Pass experience has also highlighted the need for a new capacity for critical and creative judgment that can benefit from the insertion of epistemological but also ethical-anthropological elements of literacy at different levels [31,36].

The analyses featured in this study can be also considered a seminal reflection for future issues that need further deepening, such as increasing people’s confidence in public health management, improving the relationship between policymakers and scientific and technical bodies, and increasing transparency over the use of medical data collected within the population. Previous studies have pointed out that building and maintaining trust in public health institutions, their messages and the science upon which their communication is based, is essential for improving access to, use and outcomes of public health programs [45,46]. For that matter, as shown by what has been done for robotics in Europe, in order to spread and be effective, new technologies must be able to instill confidence in future users and be reliable in performing their functions. This hinges not only on consumption but also on the real impact of technology on society [47]. In this regard, investing in transparency, which is broadly understood as providing timely information about the level of risk, communicating openly, promptly, and honestly with the public, providing evidence of what is claimed, having openness about what can be investigated, and accountability when things go wrong, is crucial to foster compliance especially with public health actions that are likely to raise concerns and emotional responses in the population, such as the adoption of the Green Pass policy [45] (p. 2). This is also in line with the aforementioned WHO document on vaccine hesitancy and the suggestion to support health officials in providing trusted and reliable information [12], consistent with the political nature of medicine [48]. Further efforts are also needed to improve communication between governments and regulatory bodies, which should be marked by prudential behavior based on the best available evidence and where this is not forthcoming, on assumptions relying on the analogy of similar phenomena.

This review was limited by the small number of articles that specifically addressed the Green Pass experience in Italy. Given the growing interest and debate raised by this policy even under future pandemic emergencies, we believe that more research will be needed on this topic and the issues highlighted by this study. Despite this limitation, we believe that our sample captured the targets under investigation and also highlighted the difficulties of extensively covering such a broad topic in a review. We decided to focus our literature search on the COVID-19 certificate in Italy as, to the best of our knowledge, studies overviewing the subject were lacking. However, this focus led us to disregard the consideration of discussing any relevant similar programs in other countries with potential advantages which may be translatable to the Green Pass in Italy. We thus believe that the results of our review could inform future exploration aimed at better understanding the specificity of the strengths, challenges, and recommendations associated with the Italian Green Pass when compared with similar experiences. Finally, Mesh Terms are not used in the search strategy, not considering, for example, possible iterations with COVID-19 mentioned in the literature. This may potentially have excluded some items addressing the targets of our study. Without ensuring the comprehensiveness of this review, however preliminary research leads us to an early summative overview of the complex experience of the Italian Green Pass.

## 5. Conclusions

Our study highlighted how the adoption of the Green Pass in Italy has contributed to coping with the COVID-19 health emergency, along with showing that this program was mostly perceived as a useful tool in pandemic management. However, the present review also identified some of its clear limitations, rooted mainly in the restrictions imposed on individuals who had not been vaccinated or recovered from COVID-19. In this context, several recommendations were suggested as an opportunity to generally rethink the role of scientific, political and health authorities and educational programs in our society, especially when it comes to affecting choices that impact health behaviors at the individual and collective level.

## Figures and Tables

**Figure 1 vaccines-10-01483-f001:**
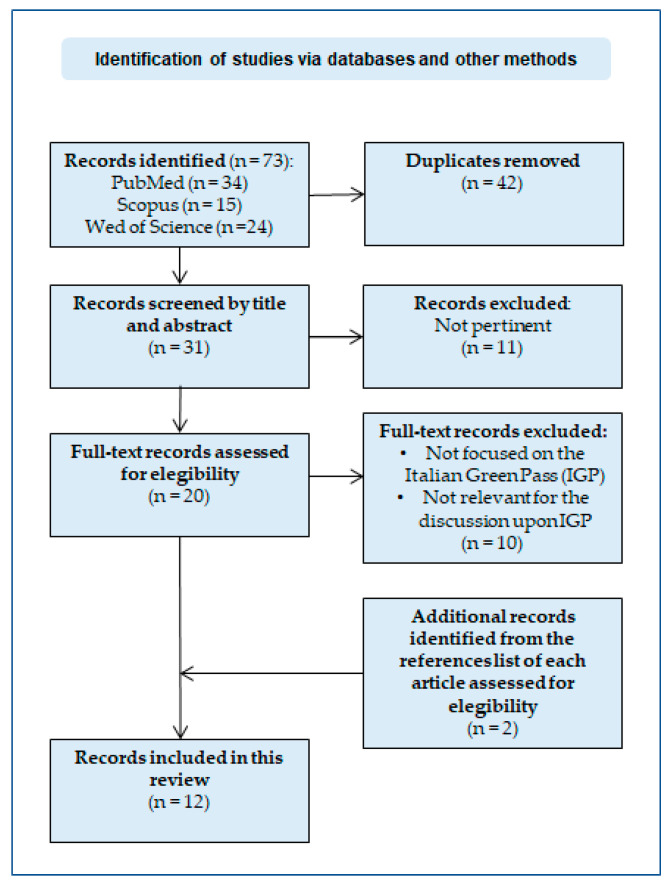
Flow diagram of the article selection process for the mini- narrative review.

**Table 1 vaccines-10-01483-t001:** Key results from selected studies.

Reference Article	Aim of the Study	Key Points about Green Pass	Proposed Recommendations
Montanari Vergallo et al. COVID-19 vaccine mandates: what are the current European public perspectives? [15].	To elaborate on the European policy choices for the prevention of SARS-CoV-2 contagion, with a close focus on the rules and regulations enacted in Italy so far.	The COVID-19 vaccination certificate refers to the EU Regulation 2021/953, of June 14, 2021, which established the Green Digital Certificate to facilitate the resumption of economic and social activities among European countries and within the Schengen area, and to stave off discriminatory practices between vaccinated and unvaccinated citizens. As for the implementation of “moral persuasion” policies, the European Union has not taken a joint decision about this Certificate, thus allowing each member state to introduce its own rules. The COVID-19 vaccination certificate has proven effective in increasing vaccination rates, but has sparked widespread protests in Europe, because it is viewed by some as a form of discrimination between those who have it, and can therefore exercise their constitutionally guaranteed freedoms, and those who do not and, therefore have such rights suspended. Such a discriminatory mechanism may not be warranted in terms of security and prevention. From the perspective of risk-benefit analysis, the vaccine has certainly been beneficial and greatly valuable for our community as a whole. Less acceptable is the policy of applying such different, uneven rules between vaccinated and unvaccinated citizens.	A strategy focused on persuasion and personal conviction is likely preferable and more in keeping with the system of safeguards for personal freedom. Only absolute clarity and straightforward information can help tackle vaccination hesitancy, in the general population and among fragile segments. Promoting public discussion between experts and citizens can go a long way towards raising awareness among the citizenry and shed light on confusing aspects. In order to deal more effectively with COVID-19 and its variants, vaccines or tests are not enough, if not coupled with a solid and clearly defined prevention strategy (such as reorganizing the health system in general and strengthening territorial medicine, intervening in the all-too-common phenomenon of overcrowded schools and classrooms, enhancing distance learning, to laying out the improvement of public transportation; to ensuring vaccines are made accessible even for countries that do not have sufficient resources to buy the drug). Policies should follow more logical and evidence-based rules. The duty of solidarity that each country has asked its citizens to fulfill, by encouraging them to become vaccinated, is ultimately insufficient.
Pamieri et al. De Jure and De Facto: An Overview on the Italian Measure in Compulsory Vaccination [22].	To trace the most important profiles of the vaccination obligation implemented *de jure* and *de facto* by the Italian government.	In August 2021, the COVID-19 Green Certification System was implemented in Italy as proof of vaccination, recovery, or a negative test result. However, since this first provision, the framework of measures in force today has evolved considerably, demonstrating an attempt to implement a growing policy of control over the spread of infection. Of the various changes made to date, the most noteworthy is the amendment made by Decree No. 172 of November 2021.20 This Decree marked the implementation of a second COVID-19 Certification, called the Green Pass Rafforzato (“Enhanced”) (or Super Green Pass), along with the existence of the Basic Green Pass. With the increase in the number of activities for which a Super Green Pass is required, the government has taken an even stronger stance on COVID-19 vaccination, imposing a sort of de facto compulsory vaccination for access to social life. Although, bys placing the word “compulsory vaccination” on a few categories of citizens, the Italian government has created a comprehensive architecture in which very few citizens can live their lives without fulfilling the, de jure or de facto, compulsory vaccination.	
Reno et al. The impact of health policies and vaccine rollout on the COVID-19 pandemic waves in Italy [29].	To describe the impact of vaccine rollout and health policies on the evolution of the COVID-19 pandemic in Italy from March 2020 to October 2021 using a set of epidemiological indicators.	In order to promote immunizations, Italy adopted a Green Pass in July 2021, and policies were developed for its implementation and use. With this certificate, people can have access to indoor restaurants, bars, theaters and other recreational venues. It is also needed to attend schools and universities. In summary, the Green Pass was implemented as a measure to encourage compliance with vaccination, increase the population’s immune coverage, and therefore reduce the circulation of the virus, allowing the “reopening” of the country. The vaccination strategy adopted as well as policies targeted to make the vaccine available (active offer of the vaccination and vaccination hubs) together with a nudging strategy (the Green Pass) had a great impact on the diffusion of the infection and the number of hospitalizations and deaths.	The study illustrated the importance of multifaceted interventions as tools to help encourage compliance with immunization.
Stefanizzi et al. Vaccination strategies between compulsion and incentives. The Italian Green Pass Experience [30].	Editorial about the Italian Green Pass experience.	The Italian Green Pass experience showed good effect of this approach in increasing immunization coverage. More than 20% of people did not have access to immunization, with a proportion of 22.6% among people aged 12–49 years and 32.8% among people aged 12–19 years.	Due to the fact that the Green Pass did not seem able to reach the last group of vaccine skeptics or no-vax people, the Italian Government now has to consider the opportunity to introduce the mandatory SARS-CoV-2 vaccination, to better ensure the health of the most vulnerable population.
Spitale et al. Concerns Around Opposition to the Green Pass in Italy: Social Listening Analysis by Using a Mixed Methods Approach [31].	To understand and describe the concerns of individuals opposed to the Green Pass in Italy, the main arguments of their discussions, and their characterization.	The declared aim of the “green passport” was to encourage citizens to receive COVID-19 vaccinations while allowing some reopening of the economy. Compared to other nudging strategies to tackle vaccine hesitancy, the Green pass appears to be a promising concept, as it incentivizes people to get vaccinated without imposing a decision. However, it has generated some debate as it can be considered a tool for discrimination based on someone’s vaccination status. The Green Pass raised an argument with regards to privacy as well: when showing their Green Pass, people are de facto obliged to disclose health information to third parties. The Green Pass has become a proxy and a catalyzer for vaccine skepticism: people generally do not argue their opposition to the Green Pass with antivaccine rhetoric but rather focus on the legal aspects and limitations of personal freedom. This opposition to the Green Pass is often justified on the grounds of a naïve idea of freedom, conceptualized in a jurisprudential, consequentialist, or deontological form.	Acknowledge the doubts of individuals opposed to the Green Pass without dismissing their opinions and arguments as ramblings. Disambiguate the purpose of the Green Pass: it should be made clear that it is a tool intended to incentivize vaccinations and thus protect people. Counteract the models of freedom in which the opposition to the Green Pass is grounded, offering alternatives. Clarify the legal basis of the Green Pass, explaining how it is founded and regulated in existing jurisprudence, and how its scope and application are defined and limited by the contingency of the pandemic. Keep informing about vaccines, with a specific focus on transparency and risk-benefit balance. The key ethical question is therefore how effective communication and management during important public health crises such as pandemics is possible without undermining privacy as a human right. Active social listening—intended as actively asking people their opinion on delicate topics such as vaccine distribution strategies or safety measures - can build trust rather than undermine it further. Engaging directly with communities by offering concerned people the possibility to voice their worries can create a sense of not only being listened to but also of being heard, recognized, and valued.
Moccia et al. Vaccine Hesitancy and the Green Digital Pass: A Study on Adherence to the Italian COVID-19 Vaccination Campaign [32].	To investigate the phenomenon of vaccination hesitancy and the underlying reasons, as well as any changes to the membership following the obligation of the Green Pass.	COVID-19 certificates should be examined within the overall broader regulatory response to the COVID-19 pandemic, which has been characterized by widespread limitations on different human rights: mobility, curfews, closure of educational institutions, and restrictions of commercial activities. The necessity for the creation of COVID-19 certificates must, therefore, be found in the need to alleviate some of the limitations placed on the general population. The COVID-19 certificate is to “facilitate safe free movement” in Europe, and it represents a tool for the regulation and governance of the pandemic, as well as for the wider governance and regulation of populations and territories, including the regulation of access to fundamental human rights. Tools, such as the Digital Green Pass, are useful for achieving overall behavioral change and effects on wellbeing, but they could harm identifiable social groups.	Trust in scientists and domestic healthcare professionals, combined with confidence in the WHO, represents an important driver of vaccine acceptance across the globe. Therefore, for trust and confidence, political leaders should assign resources to the management and communication of vaccine safety, its effectiveness, and distribution protocols to scientists and health professionals. Health professionals should, in turn, participate in developing and deploying communication strategies.
Gallè et al. Acceptance of COVID-19 Vaccination in the Elderly: A Cross-Sectional Study in Southern Italy [33].	To assess, through an online questionnaire, the acceptance of COVID-19 vaccination in a sample of older adults from southern Italy during the national vaccination campaign.	A lower vaccine acceptance was associated with the date of questionnaire fulfillment (after the mandatory implementation of the Green Pass). Mandatory measures such as the compulsory adoption of a vaccine passport may be interpreted as a threat to human rights and civil liberty, and thus decrease vaccine acceptance.	Compulsory measures must be accompanied by effective education and information strategies for the target population, paying attention to the spread of data not supported by scientific evidence. In this context, the role of reference healthcare personnel is crucial. At the same time, communication campaigns based on mass media should be tailored to those categories. Interventions supporting individuals’ autonomous motivation to get vaccinated should be programmed and implemented besides compulsory provisions of law, even in non-pandemic times. The early development of national public health laws and policies to provide a proportionate and graduated approach to compulsory vaccination in the context of a global health crisis, besides effective information campaigns, may represent an effective preparedness strategy.
Mills et al. The effect of mandatory COVID-19 certificates on vaccine uptake: synthetic -control modelling of six countries [34].	To investigate the effect of certification on vaccine uptake by designing a synthetic control model comparing 6 countries (Denmark, Israel, Italy, France, Germany, and Switzerland) that introduced certification (April–August, 2021), with 19 control countries.	For Italy, we also found a strong anticipation effect before the announcement of COVID-19 certification, followed by a decrease slightly below the average of the synthetic control group. At 30 days after implementation, daily doses in Italy were 1370 doses (1177–2421) greater than those in the synthetic control group, again suggesting a positive relationship between certification and vaccine uptake. In Italy, the youngest age group (18–24 years) had an increase in daily vaccinations directly before and after the intervention and another increase 2–3 weeks after certification. This analysis suggests that those younger than 20 years and aged 20–29 years old had increased uptake.	Although we found that certification increased vaccine uptake in certain settings and groups, COVID-19 certification alone will not increase vaccine uptake among all groups. Other measures such as geographically targeted vaccine drives or peer-to-peer and community dialogue within low-trust groups to generate understanding might be more effective for certain groups.
Oliu-Barton et al. The effect of COVID certificates on vaccine uptake, health outcomes, and the economy [35].	To estimate the effect of COVID-19 certificates on vaccine uptake for France, Germany, and Italy using counterfactuals constructed via innovation diffusion theory.	COVID-19 certificates may spur economic recovery in the short run, as newly vaccinated people can safely resume in-person economic activities, including working on-site, and consuming goods as well as services in brick-and-mortar businesses. By increasing vaccine uptake, COVID-19 certificates reduced the number of patients in ICUs and thus contributed to reducing the likelihood of stricter public measures, including lockdowns. COVID-19 certificates were associated with a sizeable, robust positive effect on vaccination rates, health outcomes, and the economy in France, Germany (albeit only significantly towards the end of 2021), and Italy. COVID-19 certificates appear to be an attractive, more inclusive alternative to vaccine mandates, focusing on the added benefits of getting vaccinated or tested rather than on punitive measures for not doing so.	The governments’ policy decisions on COVID-19 certificates should also consider additional factors, including the supply of vaccines and tests, political trust, and accessibility for marginalized groups, in order not to threaten social cohesion or exacerbate already existing inequities. International coordination and mutual acceptance of COVID-19 certificates are crucial to prevent deepening the divide between different regions.
Gometz. Green pass e discriminazione [36].	To analyse green pass certification criticism and the related conceptual and normative assumptions in order to provide an opportunity for reflection on the notion of discrimination and its criteria and conditions of employment.	The Green Pass is an effective health risk containment measure not only because it reduces opportunities for interpersonal contact between individuals who—considered in general—are at greater risk of becoming infected and infecting their neighbor, but also because it provides a formidable incentive to vaccinate of fearful citizens, the ill-informed, and the reluctant to the bitter end, as evidenced by the sudden increases in vaccination bookings recorded in Italy after the news about the adoption of the Green Pass and the extension of its mandate to most workers. Those who consider the Green Pass as intolerable discrimination, sometimes even comparing it to the membership card of the National Fascist Party or contrasting it with the yellow star or the mark on the skin of Jews during Nazism, have the burden of explaining why it is not in the same category and incite the related amount of indignant disapproval to stigmatize regulations reserving the driving of motor vehicles for those with a license, a college degree for those who pass their exams, and ownership of real estate for those who have purchased it in one of the ways established by law. The explanation cannot be based on convictions, personal beliefs, and opinions, which here, as we have seen, are totally irrelevant. Except, perhaps, to base unflattering judgments on the common sense of the bearer.	The problem of fake news in health care can possibly be addressed by the inclusion of elements of epistemological literacy in school and university curricula.
Ferraro. Passaporto vaccinale e discriminazione: stigma sociale e disuguaglianza giustificata [37].	To address accusations of state discrimination against those who are not vaccinated by choice or for health reasons related to the introduction of the COVID-19 vaccine certification in Italy.	The claim, leveled at the introduction of the Green Pass, that it constitutes a discriminatory measure against the nonvaccinated is implausible if we are talking about direct discrimination. On those who make such an accusation falls an argumentative burden: they should show that the unequal treatment in question is unjustified, taking due account of the fact that other public policy alternatives for managing and containing COVID-19 infection - or rather, the health impact of such infection - are all, at least at first glance, less palatable. If we look at the “ expressive” aspect of discrimination, related to the social stigma that the adopted measures would place on those who do not vaccinate, the opponents of the Green Pass may be right.	It is necessary to accompany the Green Pass with measures to combat marginalization, offset social injustices and inequalities, and adequately explain the good reasons supporting the vaccine passport. The proper communication of the justifications for such policy choices may help to create a context in which these choices do not express debasement and humiliation, becoming prima facie discriminatory. The accurate and truthful data on vaccine safety and efficacy is equally imperative.
Mori. Sul significato etico e filosofico del “Green Pass” o “Passaporto Vaccinale”: Un contributo alla riflessione [38].	To contribute to the current debate on the COVID-19 pandemic and its consequenc-es.	Since the strong and qualifying point of the Green Pass is vaccination, on a cultural level, the implementation of this document entails: (1) the affirmation of the concept of health as a social construct and, (2) the affirmation of the principle that when there is a dan-ger or harm to the health of others, it is permissible to restrict some freedoms.	

## Data Availability

Not applicable.

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
