# Peer review of "A Lesson from the Green Pass Experience in Italy: A Narrative Review"

_vaccines, 2022, doi:10.3390/vaccines10091483_

Round 1

Reviewer 1 Report

Estimated Authors, 

thank you for the opportunity to review this very interesting review on the Italian experience with "green pass" and associated criticisms. Authors have properly and systematically (see further about this point) collected available evidence on this specific topic, and this review - even though deliberately inconclusive, represent a significant contribution to this topic through its detailed summary of available evidence.

Still, I'm forced to share with you some criticisms that, well far from recommending the rejection of this study (I'm in fact ENDORSING the acceptance of this paper!), should recommend some improvements on it.

To begin with, I'm uncertain about the design of this paper. In fact, as you have performed a SYSTEMATIC review of available evidence on this specific topic, and several requirements from PRISMA statement have been fulfilled, I'm wondering whether an upgrade to systematic review may be acknowledged. In fact, also the supplementary table fits the requirements for a summary table (and I'm therefore recommending that it is moved from supplementary material to the main text).

Second, I think that the points your have summarized your content by should be acknowledged as a-priori search questions. Through your main text it is quite unclear why such points have been highlighted compared to other potential options. On the contrary, by stressing that heading 3.1-->3.3 would represent the outcome of your search question would avoid this potential shortcoming.

Author Response

Response to Reviewer 1 Comments

Point 1: I'm uncertain about the design of this paper. In fact, as you have performed a SYSTEMATIC review of available evidence on this specific topic, and several requirements from PRISMA statement have been fulfilled, I'm wondering whether an upgrade to systematic review may be acknowledged. In fact, also the supplementary table fits the requirements for a summary table (and I'm therefore recommending that it is moved from supplementary material to the main text). 

Response to point 1: Although I moved the summary table in the main text, as you suggested, we are apprehensive about changing the study design, toward a systematic review. The other reviewers did not make suggestions on this, and so we fear that this change may incur further criticism.

Point 2: I think that the points your have summarized your content by should be acknowledged as a-priori search questions. Through your main text it is quite unclear why such points have been highlighted compared to other potential options. On the contrary, by stressing that heading 3.1-->3.3 would represent the outcome of your search question would avoid this potential shortcoming.

Response 2: we have addeded heading 3.1-->3.3 in the introduction as targets of our review. Plese consider lines 92-95.

Many thanks for your precious suggestions.

Reviewer 2 Report

With interest, I have read the thought-provoking review by Campanozzi and coll. about the lesson learnt from the Green Pass experience in Italy. My suggestions are the following:

- adding something more on the criteria for inclusion/exclusion of the retrieved studies (particularly those belonging to field others than medicine)

- better exploit the three final questions in the conclusion section, by extensively explaining their added value in discussion, and avoid bullet points in conclusions. The sentence could be rephrase as “Additionally, the analyses featured in this study can be also considered a seminal reflection for future issues that need further deepening, such as increasing people’s confidence in public health management, improving the communication from policy makers and scientific and technical bodies, increasing transparency over the use of medical data collected within population.”

Author Response

Response to Reviewer 2 Comments

Point 1: adding something more on the criteria for inclusion/exclusion of the retrieved studies (particularly those belonging to field others than medicine). 

Response to point 1: we have added something more on the criteria for inclusion/exclusion of the retrieved studies. Please consider lines 111-113 and 118-125.

Point 2: better exploit the three final questions in the conclusion section, by extensively explaining their added value in discussion, and avoid bullet points in conclusions. The sentence could be rephrase as “Additionally, the analyses featured in this study can be also considered a seminal reflection for future issues that need further deepening, such as increasing people’s confidence in public health management, improving the communication from policy makers and scientific and technical bodies, increasing transparency over the use of medical data collected within population.”

Response to point 2: we have rephrase the sentence as you suggested and better exploit the three final questions. Please consider lines 403-421.

Many thanks for your precious suggestions.

Reviewer 3 Report

Dear editor,

Thank you for the kind invitation to review this manuscript. The article appears to be well written and structured. 

Below are my suggestion and comments on the manuscript for the authors' consideration.

Methodology

- Were any Mesh terms used for the search in pubmed

- There is probably no need to put Table 1 in the main manuscript

-> It may be better off included as a supplementary file.

- Although this is a narrative review, relevant sections from the Prisma checklist should be adopted to ensure that the manuscript covers the necessary requirements for a good narrative review.

- More details are required how search terms are derived and if any were adopted from any existing review.

- Likewise the inclusion and exclusion criteria for this study is not clear and the search period is not mentioned adequately. 

Results

- Generally well written

Discussion

- Authors may wish to consider discussing any relevant similar programs in other countries with advantages which may be translatable to the Green Pass in Italy. 

Minor point

- Please cite this relevant article in the introduction with regards to high vaccine hesitancy rates globally. 

-> https://pubmed.ncbi.nlm.nih.gov/34452026/

Some grammatical errors 

e.g. among the 12 record => it should be among the 12 records (For first sentence of results section)

Author Response

Response to Reviewer 3 Comments

Point 1: Were any Mesh terms used for the search in pubmed

Response to point 1: No use was made of Mesh terms in Pubmed.

Point 2: There is probably no need to put Table 1 in the main manuscript. It may be better off included as a supplementary file.

Response to point 2: we have put Table 1 as supplementary materials, as you suggested. Please consider lines 109-110.

Point 3: Although this is a narrative review, relevant sections from the Prisma checklist should be adopted to ensure that the manuscript covers the necessary requirements for a good narrative review.

Response to point 3: In order to meet your requests for improvement, we kindly asked for clarification regarding what relevant sections from the PRISMA checklist should be adopted to ensure that the manuscript covers the necessary requirements for a good narrative review. We followed the examples used for other narrative reviews published in vaccines. However, in trying to address your request we have added more information regarding the limitations of the study and future lines of research. We hope this works well for you. Please consider lines 380-392 and 403-421.

Point 4: More details are required how search terms are derived and if any were adopted from any existing review.

Response to point 4: we have added more details. Please consider lines 101-108.

Point 5: Likewise the inclusion and exclusion criteria for this study is not clear and the search period is not mentioned adequately.

Response to point 5: we have added more information. Please consider lines 111-113 and 118-125.

Point 6: Authors may wish to consider discussing any relevant similar programs in other countries with advantages which may be translatable to the Green Pass in Italy.

Response to point 6: We decided to focus our literature search on the Covid-19 in Italy as, to our knowledge, studies overviewing the subject were lacking. However, this focus led us to disregard to consider discussing any relevant similar programs in other countries with potential advantages which may be translatable to the Green Pass in Italy. Therefore, we have added this consideration as a limitation of the study. Please consider lines 385-392. We only added some references to similarities in the challenges raised by similar programs, as this literature was already in our possession. Please consider lines 351-362. 

Point 7: Please cite this relevant article in the introduction with regards to high vaccine hesitancy rates globally. https://pubmed.ncbi.nlm.nih.gov/34452026/

Response to point 7: we have added this article in the introduction. Please consider line 56.

Point 7: Some grammatical errors, e.g. among the 12 record => it should be among the 12 records (For first sentence of results section).

Response to point 7: we have corrected the grammatical errors.

Many thanks for your precious suggestions.

Round 2

Reviewer 3 Report

Dear Editor,

Thank you for the kind invitation to review this manuscript. I thank the authors for their efforts to revamp the manuscript. 

However, these significant issues remain which I am concerned about. 

I concur with reviewer 1 that the manuscript is structured in a similar fashion to a systematic review based on the study methodology. This was also in part my reason for bringing up the use of PRISMA statement. 

If the authors are concerned about potential criticism about the manuscript being framed as a systematic review, perhaps a scoping review could be a better way to structure the contents of the manuscript as it may be better suited for the broad concepts this review serves to cover.

-> The PRISMA-SCR checklist can be used as indicated.

Regarding the contents of which aspect of PRISMA checklist to include, I believe the authors can make the necessary adjustment depending on which route they plan to go down with regards to the study methodology.

Regardless how the article is structured above or as a narrative review, I do find it concerning that Mesh Terms are not used in the search strategy given the wealth of information it would provide and the limitations it has with regards to the content of the manuscript.

-> This is especially so given the numerous iterations how COVID-19 is termed in literature ranging from COVID-19, coronavirus 2019 etc. 

-> It would be a fundamental error / limitation in ensuring the comprehensiveness of this review, which also leads to the question if the true literature base is covered in the literature review. 

In general, conclusions from the manuscript should not contain additive information requiring citations. The authors should synthesize information based on what is already mentioned in the actual manuscript. Relevant elaborations should be made in the discussion rather than the conclusion to substantiate the points

Author Response

Response to Reviewer 3 Comments (second round)

Dear Reviewer 3,

We thank you very much for your comments.

Point 1: I concur with reviewer 1 that the manuscript is structured in a similar fashion to a systematic review based on the study methodology. This was also in part my reason for bringing up the use of PRISMA statement. If the authors are concerned about potential criticism about the manuscript being framed as a systematic review, perhaps a scoping review could be a better way to structure the contents of the manuscript as it may be better suited for the broad concepts this review serves to cover.

The PRISMA-SCR checklist can be used as indicated.

Regarding the contents of which aspect of PRISMA checklist to include, I believe the authors can make the necessary adjustment depending on which route they plan to go down with regards to the study methodology.

Response to point 1: our manuscript is structured as a narrative review which is similar to a systematic review but also different. And the differences stand in the fact that they are not systematic and follow no specified protocol, but are focused on a topic of interest. Follow an example of a narrative review already published in Vaccines Journal, that we considered in the development of our manuscript. https://doi.org/10.3390/vaccines9121477

For this reason and according to our study methodology (narrative review), we have not precisely followed the PRISMA statement that is mandatory for reporting systematic reviews, nor we have included it in our submission. This is also in line with the Journal's instructions to authors in which it is recommended to follow PRISMA guidelines for systematic reviews, not for all reviews in general. 

However, we have tried to meet your suggestions by making some adjustments as early as the first round by including more details in the methodology (specifically on inclusion/exclusion criteria, identification of keywords, time limits of the literature search,), adding study limitations, and now we have also included more information on data collection process (lines 115-116) e study limitations (lines 417-421). Therefore, without prejudice to the fact that narrative reviews are not systematic and follow no specified protocol, as some examples already published in this Journal show, and that we have already made different adjustments based on your suggestions, please if strictly necessary indicate exactly the contents of the PRISMA checklist to be integrated. 

Point 2: Regardless how the article is structured above or as a narrative review, I do find it concerning that Mesh Terms are not used in the search strategy given the wealth of information it would provide and the limitations it has with regards to the content of the manuscript.

This is especially so given the numerous iterations how COVID-19 is termed in literature ranging from COVID-19, coronavirus 2019 etc. 

It would be a fundamental error / limitation in ensuring the comprehensiveness of this review, which also leads to the question if the true literature base is covered in the literature review.  

Response to point 2: the aim of our narrative review is to provide an overview of the Italian Green pass and not to provide a systematic literature review that is comprehensive of the research on the topic. For this reason, Mesh Terms are not used in the search. Specifically, the focus of our study is not on Covid-19 while at the same time iterations of the green pass term (which is the focus of our study) were considered, such as passport and certificate. However, here again, we have tried to meet your suggestions by adding this point as a further study limitation (lines 417-421).

Point 3: In general, conclusions from the manuscript should not contain additive information requiring citations. The authors should synthesize information based on what is already mentioned in the actual manuscript. Relevant elaborations should be made in the discussion rather than the conclusion to substantiate the points.

Response to point 3: we have moved the additive information in section “Discussion” (lines 382-404), ), so the conclusions contain and synthesize information based on what is already mentioned in the actual manuscript (lines 423-431).

Round 3

Reviewer 3 Report

Dear editor,

Thank you for the invitation to review this manuscript. The authors have given some replies 

No further questions / comments from me. 

Thank you